



# Stable isotope (δ¹⁸O, δ²H) signature of river runoff, groundwater and precipitation in three river basins in the center of East European Plain

Julia Chizhova[1,2,3], Maria Kireeva[3], Ekaterina Rets[4], Alexey Ekaykin[5], Anna Kozachek[5], Arina Veres[5], Olga Zolina[6], Natalia Varentsova[3,7], Artem Gorbarenko[3], Nikita Povalyaev[8], Valentina Plotnikova[7], Timofey Samsonov[3], Maxim Kharlamov[3,4]

[1]Institute of Geography Geology of Ore Deposits, Petrography, Mineralogy and Geochemistry, Russian Academy of Science, Moscow, 119017, Russia
[2]Institute of Geography, Russian Academy of Science, Moscow, 119017, Russia
[3]Faculty of Geography, Lomonosov Moscow State University, Moscow, 119991, Russia
[4]Water Problems Institute, Russian Academy of Sciences, Moscow, 119333, Russia
[5]Arctic and Antarctic Research Institute, Climate and Environmental Research Laboratory, St. Petersburg, 199397, Russia
[6]University Josef Fourier, Grenoble, France
[7]Central administration for hydrometeorology and environmental monitoring, Moscow, 127055, Russia
[8]Voronezh State University, Voronezh, 394018, Russia

*Correspondence to*: Olga Zolina (olga.zolina@univ-grenoble-alpes.fr)

**Abstract.** Empirical study of the isotopic features of river runoff were carried out at three hydrological posts in 3 different river basins: the Zakza river in the center of East European Plane (southwest of the Moscow), the Dubna river (north of the Moscow) and the Sosna river in the south of central region. Samples of river water, groundwater and precipitation for the October 2019 - October 2021 were collected at weekly intervals. A significant supply of melted snow during spring freshet is the key factor influencing water regimes for these three river basins; varying degrees of anthropogenic flow regulation are also present. During the observation period, there were fundamentally two completely different conditions in terms of runoff formation. First, October 2019 – October 2020, there was abnormal low spring freshet  instead there was high rain flood in summer. In October 2020 – October 2021 there was a normal intra-annual flow pattern with high spring freshet. The new data of stable isotope signature of river runoff component can help to study the response of a river runoff to changing climate conditions.

## Introduction

Modern climate changes lead to a change in the behaviour of the hydrological system, an important indicator of this reaction is the isotopic characteristics of water, both on an annual and multi-year scale. Stable water isotopes are the tools for a wide range of studies in hydrology, such as determining sources of flow and pathways of water across landscapes, quantifying surface - atmosphere water fluxes and surface-groundwater interactions, and transit time, analysis origin of atmospheric moisture sources, identify the source of groundwater recharge, and study of  runoff generation and storm hydrograph (Brown



et al., 1999; McDonnell et al., 1999; Harvey and Sibray 2001; Rodgers et al., 2005; Matiatos et al. 2014; Négrel et al. 2003, Cable et al. 2011, Klaus and McDonnell, 2013; Sprenger et al. 2016, Kirchner and Allen 2020, Yang et al., 2020). Thus, stable
isotope tracers are an important tool that can be used to improve our understanding of hydrological processes and development of quantitative knowledge of water resources.

Isotope tracers can be used for local catchments with sensitive response to each precipitation event, and on a larger scale, and are highly effective means to infer the overall pattern of river runoff formation. Hydrograph separation performed by using the results of isotope analysis could show the impact of groundwater level on runoff, and help to infer features from this relation
for years with different meteorological conditions, i.e. numerically establish the ratio of the groundwater/precipitation input to runoff. In the last 5 years, anomalous hydrometeorological conditions have been observed on the East European Plain, including European Part of Russia (EPR). These conditions in turn were responsible for anomalous changes in the river regime. One example of these changes was the absence of a high spring freshet for rivers previously demonstrating this behaviour, which was later compensated by summer rain floods. Understanding of the processes governing the isotope values of
precipitation has been greatly advanced by regional and global collection networks of precipitation data (Darling et al. 2003, Kortelainen and Karhu 2004, Katsuyama et al. 2015, Yang et al., 2020), leading to the development of global and regional scale models of precipitation isotopes (Bowen 2010, Baisden et al. 2016). However, this does not apply to more eastern territories, such as European Russia. The absence of isotopic measurements of river waters, such as the GNIR project, renders these areas terra incognita. Only in recent years, regular observations of precipitation appear, which are included in the GNIP
database (Vasil'chuk et al., 2022). Nevertheless, for rivers that form the runoff of the largest water arteries of the EPR, measurements of stable water isotopes are completely absent. The Hydrometeorological Service of Russia provides regular observations of river runoff at 2092 gauging stations, these data are suitable for approximate into general runoff models; obtaining observations of isotope characteristics opens up additional opportunities for studying the precipitation-groundwater-runoff relationship. For this purpose, 3 river basins were selected: the basin of the Sosna River in the south part of central
region, the basin of the Dubna River in the north of the Moscow region and the Zakza river in the southwest of the Moscow region.

**Study area**

First of the selected rivers, the Sosna River (catchment area 16300 km$^2$), is located in in the south of the central federal district of Russia, near the Lipezk and Voronezh cities. It is a tributary of the large Don River, it has a large catchment area
and an anthropogenic impact is expressed in pollution via surface runoff, but the anthropogenic factor does not affect the flow rate. Second one, the Dubna river (catchment area 2100 km$^2$) is a river in the north of the Moscow region, it flows along the surface with an abundance of lakes, in the upper reaches of the river there is an anthropogenic factor in the runoff regulation - a hydroelectric power station reservoir, which constantly discharges water into the river. Third one, the Zakza River, is a small river (catchment area 17 km$^2$) in the south of the Moscow Region, the flow is regulated by the discharge of household water
from a residential complex. Thus, three rivers with varying degrees of anthropogenic flow regulation are represented.



The Sosna is a river in the European part of Russia, a right tributary of the Don river (Figure 1). Geomorphologically, the Sosna River Basin is a plain dissected by deep river valleys, gullies and branching ravines, characterised, by significant slopes, rapid flow, well-pronounced deep and lateral erosion. Bedrocks are represented by Devonian limestones [21]. The most grandiose limestone outcrops are located in the river valleys; The water-bearing rocks are represented by weak clayey

limestones, slightly cavernous, in some interlayers conglomerate-like, platy. Limestones alternate with marls and argillite-like clays. The depth of the first groundwater level depends on the relief and varies from 4.5 to 72.0 m, 50 - 70 m prevails. The impervious locally water-bearing Evlanovsko-Livenskaya carbonate-terrigenous suite (D3 ev-lv) is developed in the northeastern and northwestern parts of the region and is confined to the upper part of the deposits of the Upper Devonian Evlanovskaya suite. The distribution of carbonate rocks, pronounced karst formation, is easily infiltrated by precipitation.

Exploited Upper Devonian horizons and complexes in most of the territory of the region lie directly under the permeable Quaternary deposits, which ensures their aerial supply by infiltration. Vegetation is represented by broad-leaved forests, mainly in river valleys, gullies, and on watersheds and steppes. The hydrological gauge is located in Elets city. According to meteorological station in Elets city, the average monthly temperature of the warmest month (July) is 23˚C, the coldest month (January) is -5.2˚C. The winter in the region is moderately cold. The frost-free period lasts around 153 days, the longest was

209 days. The amount of precipitation in the region is 458 mm per year, 338 mm falls in the warm season. The maximum amount of precipitation falls in June and July.

Dubna is a river in the centre of the European part of Russia, in the Vladimir and Moscow regions; the right    tributary of the Volga river. Springhead, located    on the slopes of the Klin-Dmitrovskaya ridge, flows along the Upper Volga lowland. In the upper reaches, Dubna flows in a valley with steep banks, indented by the mouths of small rivers and streams. On the

Upper Volga lowland, the valley becomes wider. Geomorphologically, the basin is mainly a flat lowland with occasional low moraine hills and ridges. The sediments are represented by a thick layer of lacustrine-alluvial and fluvio-glacial deposits. Typical vegetation is pine-spruce forests with a large admixture of aspen, vast expanses of arable land with small groves of small-leaved and pine forests. The hydrological gauge on the Dubna River is located in the Verbilki village. The climate of basins of the Dubna and Zakza rivers is close to the Moscow region. In Moscow, Russia long term (1960-2020) monthly data

on precipitation amount and air temperature provide annual averages around 702 mm and 5.6°C respectively. Long term temperature of the coldest month (January) is –7.9°C and the warmest (July) is 19°C. Half of the annual precipitation amount falls from June to October.

The Zakza River flows through a hilly, strongly rugged plain, typical landscapes are arable and forested. The right slope of the valley is steep, sometimes up to 15-20 m high, dissected by ravines. The vegetation is represented by mixed broad-

leaved forests and shrubs. The river bed is winding, gravelly-sandy, slightly deformable. In severe winters, the river does not freeze, the ice formation is unstable, and there are a large number of polynyas. The geological structure of sediments is represented by interbedding of upper moraine loams, on which there are supra-morainic sands of different thickness within the catchment area - from 1.5 to 7-8 m. The artesian waters of the Podolsk-Myachkovsky and Aleksinsko-Protvinsky aquifers lie



at a depth of more than 80 meters. The hydrological gauge is located in the village of Bolshoye Sareevo. The depth of the
aquifer on the territory of the Bolshoye Sareevo village was determined from real wells from 83 to 95 meters.

## Methods and results

### Sampling and analytical procedures

Samples of river water, groundwater and precipitation from September 2019 to October 2021 were collected at weekly
intervals at 3 hydrological gauges (Figure. 1). River water and groundwater were sampled in the morning hours, at 10 AM;
precipitation samples represent an integral sample of all precipitation that fell in the week preceding the date of sampling.
Daily precipitation samples were collected with unheated precipitation collectors O-1 (Tretyakov rain gauge) installed at a
height of 2 meters from the ground. After every precipitation event this portion was poured into a polyethylene canister for
storage. During the week each new portion of daily precipitation was added to it. At the end of the week, an integral sample
from the canister was poured into a polyethylene 10 ml tube. Samples of River water were collected by hand using sampling
tube along a vertical shaft, which was lowered into the water. Groundwater at Dubna river gauge at Verbilki was sampled from
a well with a depth of 10 m, at Sosna (Elets) and Zakza (Bolshoe Sareevo) from a deep well (80 m depth) for water supply.
The samples were not filtered, stored in polyethylene 10 ml tubes, sealed with paraffin tape to protect samples from
evaporation.
Isotope analysis was carried out at the Climate and environmental research laboratory in Arctic and Antarctic Research Institute
using a Picarro L2130-i isotope analyzer. The accuracy was 0.04 ‰ for the $\delta^{18}$O value measurements and 0.5 for the $\delta^2$H value
measurements. The values were calibrated in the VSMOW-VSLAP scale. The isotopic abundances of $^{18}$O and $^2$H are reported
using the δ notation relative to the IAEA standard Vienna Standard Ocean Water (VSMOW) following Eq. (1):

$$\delta = \left(\frac{R_{sample}}{R_{VSMOW}} - 1\right) \times 1000\ ‰ , \qquad (1)$$

where R is the ratio of the heavier isotope relative to the lighter isotope (i.e., $^{18}$O/$^{16}$O or $^2$H/$^1$H). We have used several
international and commercially available standards to validate our isotope measurements and to report them relative to the
VSMOW-SLAP scale of the International Atomic Energy Agency IAEA. The Picarro L2130-i isotope analyzers were
calibrated with at least two International Atomic Energy Agency (IAEA) standards (VSMOW2 and SLAP2), so that isotope
measurements were comparable across laboratories and instruments. Every measure sequence was calibrated by measuring
three laboratory reference standards every 20 to 25 samples. For each sequence, the average values of these standard isotope
measurements were used to obtain the linear calibration equation and to determine instrument drift.

### Data Records

Runoff average values of $\delta^{18}$O and $\delta^2$H at Sosna gouge for the entire observation period were −10.95 ‰ (from −8.86
to −12.48 ‰) and −81.3 ‰ (from −74.5 to −91.4 ‰), respectively (Figure 2). In the 2019-2020 hydrological year, there was



no period of high spring water, which was caused by unusual meteorological conditions in winter. The low $\delta^{18}O$ values of river

runoff were associated with the direct input of autumn and winter precipitation into the river flow. In the summer of 2020, the

$\delta^{18}O$ values increased due to the participation of summer rains. In 2021, a decrease in $\delta^{18}O$ of the river runoff was noted during

high spring water due to the inflow of snowmelt. Groundwater average values of $\delta^{18}O$ and $\delta^2H$ were −11.51 ‰ (from −11.04

to −11.85 ‰) and −84.3 ‰ (from −81.13 to −86.27 ‰), respectively. These values indicate that groundwater recharge in the

Sosna catchment was supplied by an increased amount of flood waters, which have lower $\delta^{18}O$ values compared to the average

annual values, since they are associated with the melting of snow cover. This corresponds to precipitation fallen in December,

January and February, which provides up to 50% of the annual recharge of the aquifer. The small amplitude of $\delta^{18}O$ values of

groundwater is associated with significant averaging of the precipitation isotope signal, which occurs when water stays in the

aquifer for a long time. Local atmospheric precipitation has a range of $\delta^{18}O$ and $\delta^2H$ values from 1.99 to −18.71 ‰ and from

−11 to −142.4 ‰, respectively, following a distinct seasonal pattern with heavier isotopes in summer and lighter isotopes in

winter.

The local meteoric water line (LMWL), determined by least-squares regression is $\delta^2H = 6.32 \times \delta^{18}O - 12.68$, $R^2 =$

0,95, which reflects the evaporation of rain during the summer months, most likely due to the more southerly position of the

basin. A linear $\delta^{18}O$-$\delta^2H$ regression based on isotope data of river runoff also indicate the processes of evaporation of river

water (Figure 3), or input of evaporated rain to river flow.

Runoff average values of $\delta^{18}O$ and $\delta^2H$ at Dubna gouge for the entire observation period were −10.36 ‰ (from −8.56

to −12.88 ‰) and −78.8 ‰ (from −65.9 to −95.3 ‰), respectively. The high $\delta^{18}O$ values of river runoff were noted for the

beginning of summer 2020, when the flood was formed by heavy summer rains. The low $\delta^{18}O$ values of river runoff were

associated with spring high water in 2021. On a long-term scale, Dubna river is characterised by a significant supply of melted

snow during spring high water. In June 2020, the rain flood significantly exceeded the spring high water, which was associated

with a snowless winter of 2019-2020.

Groundwater average values of $\delta^{18}O$ and $\delta^2H$ were −11.48 ‰ (from −9.89 to −14.53 ‰) and -86.4 ‰ (from −79.3 to −110.7

‰), respectively. The lowest $\delta^{18}O$ value = −14.53 ‰ obtained once on 05/05/2021, perhaps, represented an accidental hit of

the isotope signal of snow melt into groundwater. Excluding this event, the average $\delta^{18}O$ value was −11.49 ‰. Obviously, the

average values of $\delta^{18}O$ and $\delta^2H$ of groundwater are lower than isotope values of river waters. This indicates a relatively greater

contribution  the infiltration of winter/spring precipitation to groundwater recharge in compare to annual precipitation. The

pronounced difference between the $\delta^{18}O$ values of river runoff and the $\delta^{18}O$ values of groundwater throughout the year (Figure

4) indicates that the river runoff has always had an isotopically heavy component and during the observation period there were

no conditions when the river runoff was completely supplied by groundwater. Besides, the seasonal groundwater isotope

amplitude is large, which indicates a direct relationship with infiltration of precipitation without temporary retention in the

aquifer.

Local atmospheric precipitation has a range of $\delta^{18}O$ and $\delta^2H$ variations from −3.49 to −17.51 ‰ and from −21 to −129.5 ‰,

respectively. Precipitation isotope signatures follow a distinct seasonal pattern with higher $\delta^{18}O$ values in summer and lower



values in winter. The seasonal runoff and groundwater isotope amplitude is damped compared to that of precipitation due to mixing with water storage in catchment. The local meteoric water line (LMWL), determined by least-squares regression is $\delta^2H = 7.62 \times \delta^{18}O + 3.45$, $R^2 = 0{,}98$, which closely follows the Global Meteoric Water Line (GMWL, $\delta^2H = 8 \times \delta^{18}O + 10$).

Runoff average values of $\delta^{18}O$ and $\delta^2H$ at Zakza gouge for the entire observation period were -10.76 ‰ (from -6.94 to -13.6 ‰) and -81.92 ‰ (from -59 to -101 ‰), respectively. In the 2019-2020 hydrological year, there was no period of high spring water, which was caused by unusual meteorological conditions in winter, but a big flood in June 2020 due to intensive rains (Figure 4). The high $\delta^{18}O$ values of river runoff were associated with summer rains in 2020. The low $\delta^{18}O$ value of the river runoff was noted during high spring water in 2021 caused by snowmelt.

Groundwater average values of $\delta^{18}O$ and $\delta^2H$ for the entire observation period were $-11.48$ ‰ (from $-8.61$ to $-12.22$ ‰) and -86.3 ‰ (from $-77.7$ to $-89.1$ ‰), respectively. On the $\delta^{18}O$-$\delta^2H$ plot there is a clear difference between isotope signature of groundwater during 2019-2020 and 2021. We attribute this effect to the inflow of wastewater from the residential complex, in which new wells for water supply were put into operation. These waters enter the Zakza River and significantly determine the isotopic parameters of its runoff. Starting from January 2021, the $\delta^{18}O$ values of groundwater are very uniform from $-11.8$ to $-12.22$ ‰, indicating that this is water from a deep aquifer, where the water stays for a long time and in which seasonal variations of precipitation are homogenised. Against this background of such a homogeneous isotope signal of groundwater, the $\delta^{18}O$ variations of runoff, associated with the direct contribution of precipitation are more noticeable.

Local atmospheric precipitation has a range of $\delta^{18}O$ and $\delta^2H$ values from $-3.15$ to $-28.54$ ‰ and from $-23$ to $-221.8$ ‰, respectively, following a distinct seasonal pattern with heavier isotopes in summer and lighter isotopes in winter. The local meteoric water line (LMWL), determined by least-squares regression is $\delta^2H = 7.91 \times \delta^{18}O + 3.5$, $R^2 = 0.98$, which closely follows the Global Meteoric Water Line.

**Data availability**

The presented datasets are available open access via the PANGAEA repository (Chizhova et al., 2022 https://doi.pangaea.de/10.1594/PANGAEA.942291)

The river discharge is measured by the Federal State Budgetary Institution "Central Administration for Hydrometeorology and Environmental Monitoring" (FGBU "Central UGMS") (URL: httpswww.cugms.ru, accessed 20 March 2022). These data cannot be provided with our dataset due to legal restrictions; however, they can be requested free of charge from hydrology department (URL: https www.cugms.ru; Dubna river at Verbilki - #75079; Zakza river at Bol.Sareevo #75438; Sosna river at Elets #78054). Alternatively, data on the water level of Dubna and Sosna rivers at corresponding gouges are available via open access database https://allrivers.info (https://allrivers.info/gauge/dubna-verbilki?; https://allrivers.info/gauge/byistraya-sosna-elec ).



**Author contributions**

J.Ch., K.R. and M.K. designed the study, N.P. and V.P. collected samples of river runoff and groundwater, O.Z. developed
the study of precipitation, A.E., A.K. and A.V. performed the isotope measurements, N.V., T.S. and M.Ch. provided runoff
hydrological data, J.Ch. prepared the paper with contributions from all authors.

**Acknowledgements**

The work was supported by the Russian Science Foundation, project 19-77-10032 (sampling of river water, groundwater,
isotope analysis), concept and organization of the study of atmospheric precipitation supported by the Megagrant project
(agreement № 075-15-2021-599, 8.06.2021).

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

Table 1

| River (Gouge) | Latitude, longitude | Basin area, km$^2$ | Average catchment height, m | Observation period | Number of river runoff samples | Number of groundwater samples | Number of precipitation samples |
|---|---|---|---|---|---|---|---|
| Dubna (Verbilki) | 56.53 N, 37.6 E. | 2100 | 179 | 02.10.2019-31.10.2021 | 109 | 107 | 64 |
| Sosna (Elets) | 52.62 N. 38.47 E. | 16300 | 156 | 01.09.2019-01.10.2021 | 108 | 54 | 27 |
| Zakza (Bolshoe Sareevo) | 55.71 N, 37.18 E. | 17 | 191 | 03.10.2019-31.11.2021 | 115 | 114 | 103 |





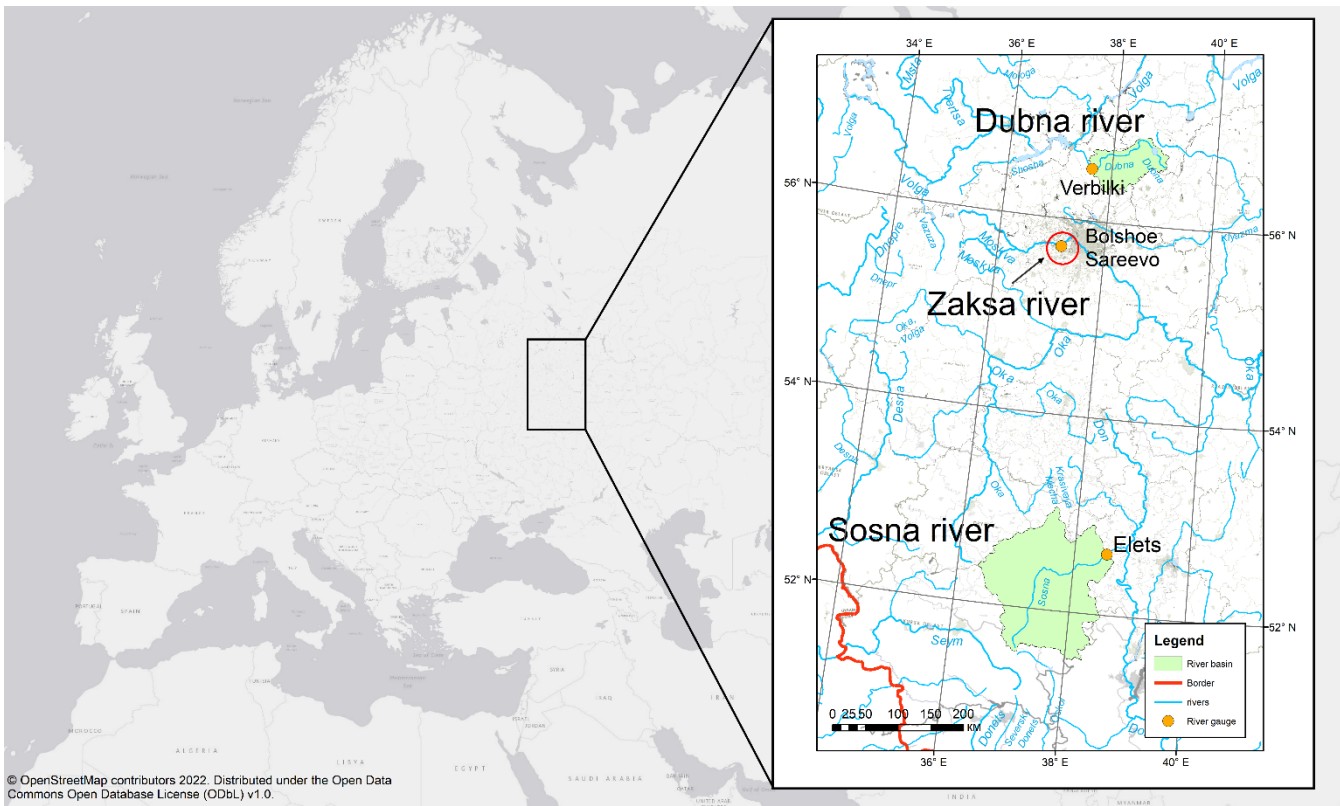

**Figure 1: Location of three river basins in the centre of the East European Plain.**

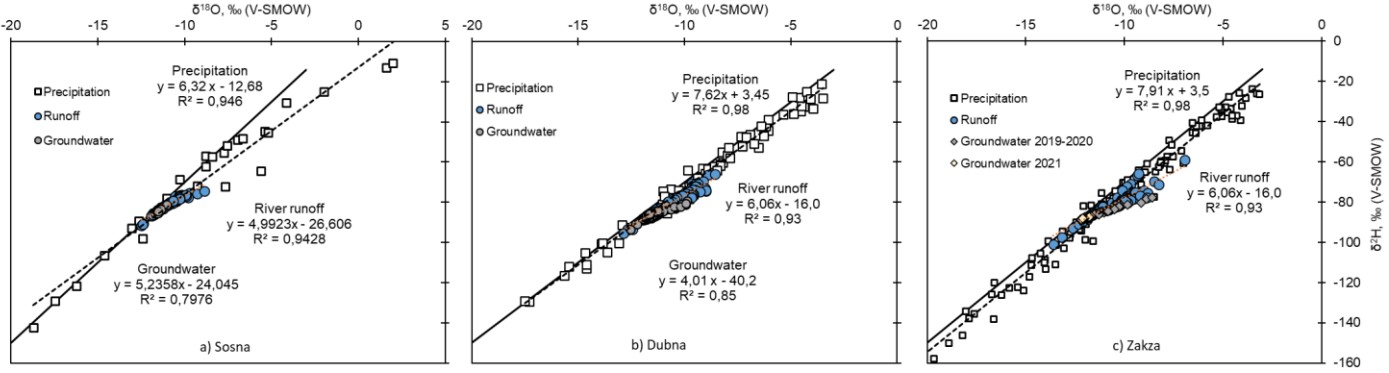


**Figure 2: Runoff of Sosna river at the Elets gauging station and δ¹⁸O values of river water, groundwater and precipitation for the period from 01.09.2019 to 01.10.2021.**

**Figure 3: The δ²H vs δ¹⁸O plot of river runoff, groundwater and precipitation at gauging station of the a) Sosna river basin, b) Dubna river basin, c) Zakza river basin.**





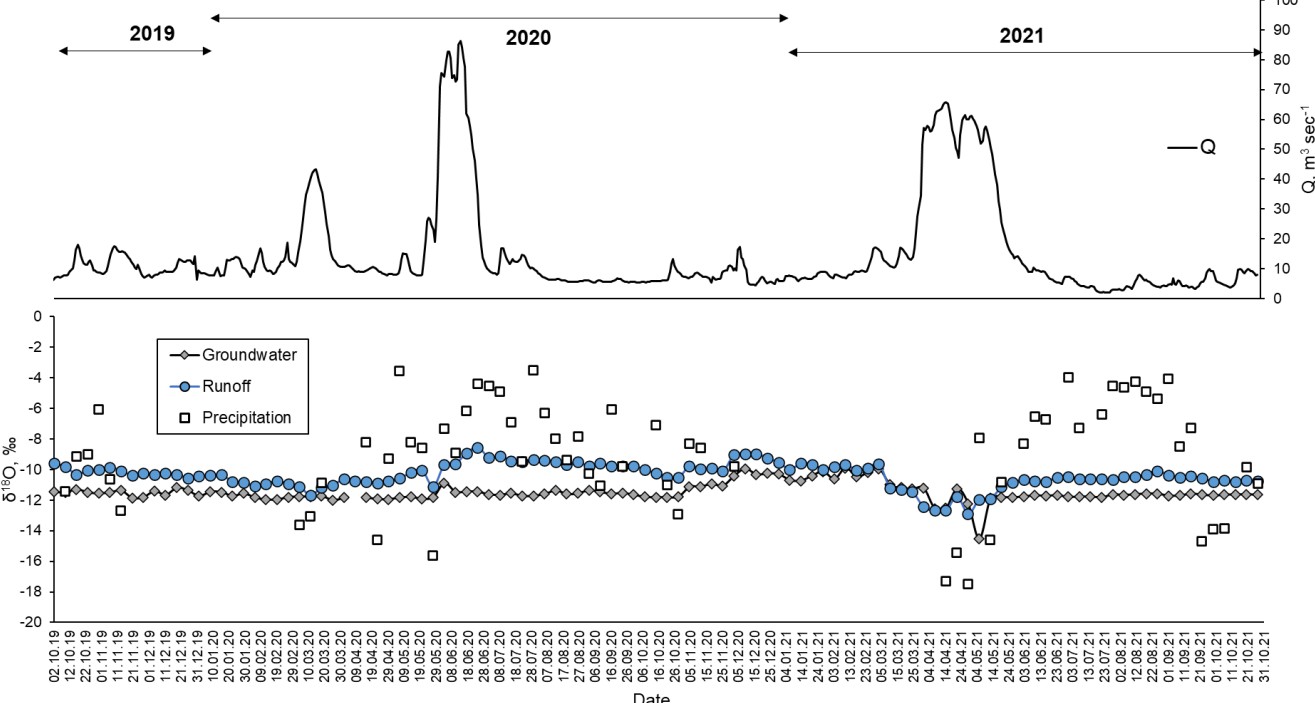

**Figure 4: Runoff of Dubna river at the Verbilki gauging station and δ$^{18}$O values of river water, groundwater and precipitation for the period from 01.10.2019 to 31.10.2021.**





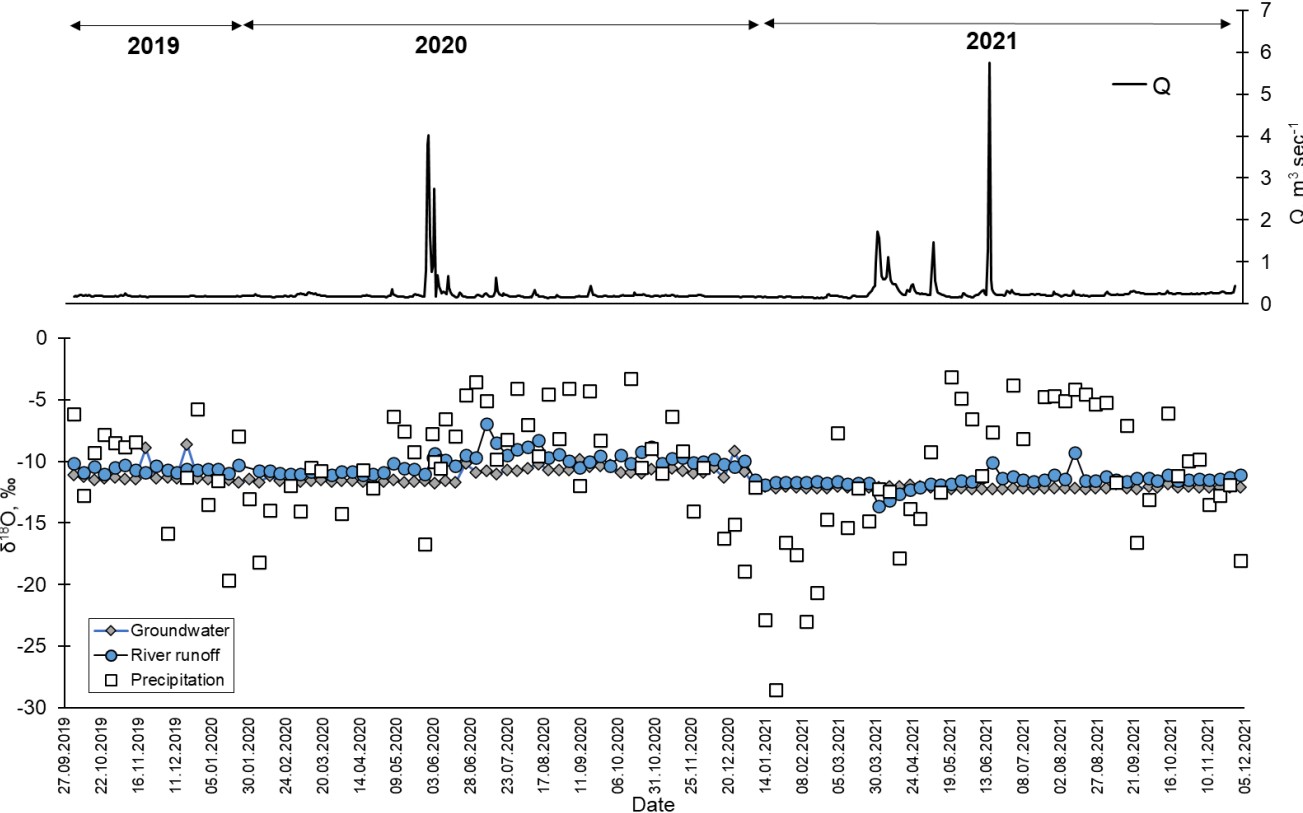

**Figure 5: Runoff of Zakza river at the Bolshoe Sareevo gauging station and δ¹⁸O values of river water, groundwater and precipitation for the period from 01.10.2019 to 31.11.2021.**
