# Peer review of "Stable isotope ( $\delta^{18}$ O, $\delta^{2}$ H) signature of river runoff, groundwater and precipitation in three river basins in the center of East European Plain"

_Earth System Science Data, 2022_

## Referee Comment (RC1)

Comments to "Stable isotope ($\delta^{18}$O, $\delta^2$H) signature of river runoff, groundwater and precipitation in three river basins in the center of East European Plain" by Chizhova et al.,

The authors collected samples for isotopic analysis in three basins with collecting areas varying from 17 km$^2$ to 16300 km$^2$ in eastern Europe. The samples and data are valuable to the isotope community, and the study falls into the scope of the journal.

Major concerns:

a) The descriptions of the importance of the data needs to be strengthened, e.g., readers will not know how many datasets were collected and analyzed in the study from the 'Abstract' section; also, the scarcity, including the high sampling frequency and completeness, of the data should be described more clearly.

b) The writing should be checked by native English speakers, and the grammatic errors could be found throughout the manuscript.

Monir concerns:

Line 18: '3' should be replaced by 'three' here.

Line 19: 'during October 2019'

Line 21: 'in' there three basins

Line 22: 'fundamentally' should be deleted

Line 28: 'Recent climate change…'; 'and an important ….'

Line 29: are effective tools for ….

Line 31: surface – atmosphere water 'exchanges' and …

Lines 32-34: These references should be divided into several sections according to which description they supported.

Line 40: i.e. quantitatively establish…

Lines 48-49: Full names should be shown for 'GNIR' and 'GNIP' when they appeared for the first time.

Above minor concerns are only examples, and the entire manuscript should be carefully checked.

---

## Community Comment (CC2)

*Chizova et la present an extensive dataset of O and H stable isotope ratios in precipitation, ground and surface (river) waters from the East European Plains in Russia. This is most welcome for the wider stable isotope, hydrology and paleoclimate scientific communities as it would help expand the spatial and temporal coverage of existing datasets towards region that is poorly covered by such data. I suggest publication of the ms and associated data. A few minor comments and one major concern should be nevertheless addressed:*

1. *The extremely low intercept of the Sosna LMWL suggest something is (potentially) wrong with the data. Checking it at the link provided by the authors (Pangaea) shows several months (1/3 of the data points) with very to extremely low d-excess values (as low as -32 ‰). The samples have been collected mostly in summer, but some are also from April and May. While sub-cloud evaporation could cause low d-excess, the values reported here are too low, hinting at post-deposition processes (e.g., improper handling of samples after collection) and/or problems during analysis (e.g., contamination with volatile organics, a problem that would affect samples analyzed using CRDS systems – as done by the authors – contrary to analyses performed using IRMS analyzers). Some of the precipitation samples collected at Zakza and Dubna also display low (negative) d-excess values, but within the "normal" range of samples affected by post-depositional evaporation. As groundwater and river samples do not show similarly extreme d-excess values, I believe the issue stems from improper handling of the samples collected at Sosna. Please check (and if samples are still available) re-run them on an IRMS. Else, a cautionary note should be inserted in the main text of the manuscript.*

We agree with Dr. Aurel Perşoiu and appreciate his comments on our data. Indeed, for several precipitation samples collected in the summer of 2019, 2020 and 2021, extremely low values of d-excess were obtained. As a rule, negative d-excess values in precipitation are associated with sub-cloud evaporation until raindrops reach the ground. The highly negative d-excess values (lower than −10‰) have been reported for samples 28.06.2020; 6.06.2021; 13.06.2021; 4.07.2021; 25.07.2021. These samples represent thunderstorms and strong showers precipitation events. We cannot state that this is just sub-cloud evaporation effect or intracloud evaporation associated with vertical ascending air flows in a thundercloud. This could be the evaporation of the sample during its storage and transport. Thereby these samples cannot be considered as completely reliable. However, they can explain the effects of evaporation trend on $\delta 18O$-$\delta 2H$ diagram for river water as a reaction to the addition of such low d-excess precipitation to the river (see Fig. 2).

We do not link these data to analysis problems. Isotope analysis was carried out at the Climate and environmental research laboratory in Arctic and Antarctic Research Institute. This laboratory specializes in isotope analysis of Antarctic ice cores, high accuracy of measurements is confirmed by regular Inter-laboratory Comparison (WICO) by IAEA. All measurements were made according to repeatedly tested protocols. In addition, samples with these low d-excess values were reanalyzed 3 times on different days. We see no possibility to carry out an additional re-run of these samples by the IRMS method. Even though IRMS is the "gold standard" of measurements, at Picarro we get high accuracy data. In addition, the samples were analyzed 1 year ago, and all this time they were kept in the archive. Perhaps reanalyzing them will create more questions than answers. We added to the text the sentence that the isotope composition ($\delta 18O$, $\delta 2H$) of these samples could be changed by evaporation during storage and transportation and these samples cannot be considered as completely reliable.

2. *To make full usage of the data, precipitation amount and air temperature data should be made available.*

We added a new data on precipitation amount and air temperature, weighted by precipitation amount to our dataset, it could be access via https://doi.org/10.1594/PANGAEA.948718

3. *A note on nomenclature. Please check the ms for improper usage of stable isotope jargon (e.g., "stable water isotopes" – water has no stable isotopes per se, only oxygen and hydrogen in water have etc)*

Checked and corrected

4. *Sentence structure and grammar: please check (overall) the structure of sentences, to often these are very long so that by the end of a sentence the reader loses the information at the beginning*

The text has been checked and edited. Also, the manuscript was read and corrected by native English speaker.

We thank Dr. Aurel Perşoiu for comments and suggestions.

---

## Author Comment (AC1)

*Comments to "Stable isotope (δ 18O, δ 2H) signature of river runoff, groundwater and precipitation in three river basins in the center of East European Plain" by Chizhova et al., The authors collected samples for isotopic analysis in three basins with collecting areas varying from 17 km2 to 16300 km2 in eastern Europe. The samples and data are valuable to the isotope community, and the study falls into the scope of the journal.*

*Major concerns:*

*a) The descriptions of the importance of the data needs to be strengthened, e.g., readers will not know how many datasets were collected and analyzed in the study from the 'Abstract' section; also, the scarcity, including the high sampling frequency and completeness, of the data should be described more clearly.*

We have made some correction to 'Abstract' and 'Introduction' sections to show the importance, relevance and completeness of our data.

*b) The writing should be checked by native English speakers, and the grammatic errors could be found throughout the manuscript.*

The manuscript was read and corrected by native English speaker.

Monir concerns:

*Line 18: '3' should be replaced by 'three' here. Line 19: 'during October 2019'*

Corrected

*Line 21: 'in' there three basins*

Corrected

*Line 22: 'fundamentally' should be deleted*

Corrected

*Line 28: 'Recent climate change…'; 'and an important ….'*

Corrected

*Line 29: are effective tools for ….*

Corrected

*Line 31: surface – atmosphere water 'exchanges' and …*

Corrected

*Lines 32-34: These references should be divided into several sections according to which description they supported. Line 40: i.e. quantitatively establish…*

Corrected

*Lines 48-49: Full names should be shown for 'GNIR' and 'GNIP' when they appeared for the first time. Above minor concerns are only examples, and the entire manuscript should be carefully checked*

Corrected

We are very grateful to the Reviewer for comments and suggestions.